# Efficacy and safety profile of combining antiangiogenic agents with chemotherapy in patients with advanced malignant pleural mesothelioma: A systematic review and meta-analysis of randomized controlled trials

**Wei Tian[1], Qian Guo[2], Daidi Fu[1], Xiao Ma[3], Rui Wang[1] \***

**1** Department of Oncology, Zibo Central Hospital, Shandong University, Zibo, Shandong, Peoples' Republic of China, **2** Day Care Unit, Zibo Central Hospital, Shandong University, Zibo, Shandong, Peoples' Republic of China, **3** Department of Internal Medicine, Zhangqiu People's Hospital, Zhangqiu, Shandong, People' Republic of China

\* wangrui2023@126.com

## Abstract

### Objectives

Several prospective trials had been reported on chemotherapy with or without antiangiogenic agents in patients with advanced malignant pleural mesothelioma (MPM), with diverse results. We performed this systematic review and meta-analysis to evaluate the efficacy and safety of the combination regimen.

### Methods

We systematically identified trials in several databases, including MEDLINE, EMBASE, Cochrane Central Register of Controlled Trials, ASCO Abstracts and ESMO Abstracts. All the randomized controlled trials (RCTs) about chemotherapy combined with antiangiogenic agents in advanced MPM were identified. Overall survival (OS) was the primary outcome, while progression-free survival (PFS), overall response rate (ORR) and serious toxicities were the secondary outcomes. Review Manager 5.3 was used to perform the statistical analyses. Stata 12.0 was used to assess the publication bias of egger's test.

### Results

5 randomized controlled trials containing 1250 patients were finally included in this analysis. Statistical analyses showed that the addition of antiangiogenic agents to chemotherapy could prolong OS [HR 0.79 (0.71–0.89), p<0.0001] and PFS [HR 0.75 (0.68–0.84), p<0.00001] in advanced MPM, especially in the epithelioid subgroup, with a tolerable toxicity profile. No significant difference was found in the analysis of ORR [HR 1.13 (0.95–1.35), p = 0.18]. Heterogeneity was found in the analyses of PFS and ORR, which might be caused by the limitation in uniform evaluation of tumor response.

**Data Availability Statement:** All relevant data are within the paper and its Supporting Information files.

**Funding:** The authors received no specific funding for this work.

**Competing interests:** The authors have declared that no competing interests exist.

## Conclusions

The combination of antiangiogenic agents with chemotherapy showed superior over chemotherapy alone in patients with advanced MPM. More prospective trials should be warranted to identify patients who would most likely benefit from the combination regimen.

## Introduction

Mesothelioma is a rare but aggressive cancer, with 30,870 new cases and 26,278 deaths worldwide in 2020 [1]. Malignant pleural mesothelioma (MPM), the most common form of malignant mesothelioma, is usually diagnosed at the advanced stage. MPM is classified into 3 main histologic subtypes: epithelioid, sarcomatoid and biphasic [2]. For patients with unresectable disease, chemotherapy of pemetrexed combined with platinum is the basic treatment option [3]. Although regimen of pemetrexed plus cisplatin brings clinical benefits compared with cisplatin alone, the median overall survival (OS) is only 12.1 months [4].

Angiogenesis plays an important role in cancer progression, which has become an important target in cancer therapy [5]. Vascular endothelial growth factor (VEGF), fibroblast growth factor (FGF), platelet-derived growth factor (PDGF) and their corresponding receptors regulate the signaling pathways of cancer angiogenesis. Signaling pathways involved in angiogenesis are thought to be associated with the development of MPM [2]. The VEGF serum levels are verified to be higher in patients with MPM [6]. VEGF receptor 2, FGF receptor 1, FGF2, and FGF18 have been found to be overexpressed in mesothelioma cells [7, 8]. PDGF also plays a role as autocrine growth stimulating factor in the development of malignant mesothelioma [9]. Several studies have investigated antiangiogenic agents such as cediranib, sorafenib and sunitinib in the second-line setting when used as monotherapy in advanced MPM. However, the results did not demonstrate sufficient clinical benefits in these studies [10–12].

In the Mesothelioma Avastin Cisplatin Study, bevacizumab combined with pemetrexed and cisplatin significantly improved the OS compared with pemetrexed and cisplatin alone, which indicated that addition of antiangiogenic agents to standard chemotherapy might be a rational approach in MPM [13]. Over the past several years, a number of randomized controlled trials (RCTs) have evaluated the efficacy of adding antiangiogenic agents to standard chemotherapy in patients with advanced MPM, but with diverse results. Therefore, we performed this systematic review and meta-analysis to compare the antiangiogenic agents plus standard chemotherapy versus chemotherapy alone for such patients.

## Materials and methods

### Search strategy

We collected all the relevant studies by searching MEDLINE, EMBASE, Cochrane Central Register of Controlled Trials, ASCO Abstracts and ESMO Abstracts up to June 2023. In order to identify all the RCTs in MEDLINE (Ovid format), the Cochrane highly sensitive search strategy was used. **Table 1** showed the search strategy used in this analysis.

### Inclusion and exclusion criteria

The criteria for inclusion in this study were as follows: (1) Type of patients: adults ($\geq$18 years) had a histologically proven diagnosis of malignant pleural mesothelioma with locally advanced or metastatic disease; (2) Type of study: phase II or phase III RCTs comparing antiangiogenic

**Table 1. Search strategy for MEDLINE (Ovid format) used in this meta-analysis.**

| | |
|---|---|
| 1. randomized controlled trial.pt. | 18. antineoplastic$.tw. |
| 2. controlled clinical trial.pt. | 19. or/15-18 |
| 3. randomized.ab. | 20. aflibercept |
| 4. placebo.ab. | 21. sorafenib |
| 5. drug therapy.fs. | 22. sunitinib |
| 6. randomly.ab. | 23. axitinib |
| 7. trial.ab. | 24. apatinib |
| 8. groups.ab. | 25. vandetanib |
| 9. or/1-8 | 26. nintedanib |
| 10. humans.sh. | 27. pazopanib |
| 11. 9 and 10 | 28. cediranib |
| 12. exp pleural mesothelioma/ | 29. bevacizumab |
| 13. (pleural adj5 mesothelioma$).mp. | 30. ramcirumab |
| 14. or/12-13 | 31. or/20-30 |
| 15. exp drug therapy/ | 32. 14 and 19 |
| 16. chemothera$.tw. | 33. 31 and 32 |
| 17. drug therap$.tw. | 34. 11 and 33 |

We identified all the randomized controlled trials about chemotherapy combined with angiogenesis agents in advanced MPM. All references of the included studies were scanned manually to identify relevant trials. There was no language restriction in the search.

agents plus standard chemotherapy versus chemotherapy alone in first-line or second-line treatment; (3) There was at least one endpoint about OS or progression free survival (PFS) reported in the study; (4) If more than one publication showed data for the same study, only the most recent publication was included.

The criteria for exclusion were as follows: (1) Quasi-randomized trails which had a greater potential for bias were excluded. (2) Cross-over studies which had a significant impact on the overall effect of OS were also excluded.

Two reviewers (Wei Tian and Rui Wang) independently screened all the details of each study to confirm it met the inclusion criteria. When discrepancies arose, we resolved them through discussing with a third reviewer (Daidi Fu).

## Data extraction

Data were extracted from each included study by two reviewers (Wei Tian and Rui Wang) independently. Discrepancies and uncertainties were resolved by consensus with a third reviewer (Daidi Fu). OS was the primary endpoint, defined as the time from randomization to death from any cause. PFS was one of the secondary endpoints, defined as the time from randomization to disease progression or death. Other secondary endpoints included overall response rate (ORR) and serious toxicities. ORR was defined as the proportion of participants achieving confirmed complete response (CR) or partial response (PR). The hazard ratio (HR) and 95% confidence interval (CI) of time-to-event data including OS and PFS were synthesized and analyzed, as provided by Jayne F Tierney et al [14]. Risk ratio (RR) and 95% confidence interval of dichotomous data for ORR and toxicities were also analyzed. We extracted the HRs from the included studies or synthesized them from the reported events and the corresponding p-value of the log-rank statistics. When the above-mentioned data were not reported, we would digitize the Kaplan Meier curve and use Guyot's method to calculate the

summary estimate [15]. To assess the methodological quality of the 5 included studies, Jadad score method was adopted. The Jadad score, which ranged from 0 to 5, was based on three items, including descriptions of randomization, double blindness and withdrawals. When mentioning randomization, double blindness, and withdrawals, each study would be assigned a score of 1 point. If the appropriate methods of randomization or double blindness were described, an additional 1 point would be awarded. Studies scoring Jadad $\geq$ 3 were classified as high quality.

## Statistical analyses

Statistical analyses were performed using Review Manager 5.3 software. Time-to-event data were analyzed using HR. Dichotomous data were analyzed using RR. Statistical heterogeneity between studies was calculated by the chi-square test and expressed with the $I^2$ index. For the $I^2$ statistic, a $I^2$ value higher than 50% meant great heterogeneity, 25%-50% meant moderate heterogeneity, less than 25% meant low heterogeneity. Fixed-effect model (Mantel-Haenszel method) was used if no significant heterogeneity was identified. In the absence of significant heterogeneity ($p<0.1$, or $I^2>50\%$), random-effect model or sensitivity analysis would be used. In addition, Egger's test performed by stata 12.0 was also adopted to assess the possibility of publication bias. A two-sided $p<0.05$ was considered statistically significant.

## Results

### Study identification

528 records were rigorously screened in our search progress. Finally, 5 publications related to 5 RCTs (1250 patients) that compared the combination of antiangiogenic agents and chemotherapy with chemotherapy alone were included in the analysis [13, 16–19]. Three phase II [17–19] and two phase III [13, 16] trials were included. One trial reported the results of bevacizumab combined with chemotherapy [13]. Two trials compared nintedanib and chemotherapy with chemotherapy alone [16, 19]. One trial was designed to illustrate the efficacy of the addition of cediranib to chemotherapy [17]. One trial was about ramucirumab combined with chemotherapy [18]. **Fig 1** showed a flowchart for identifying relevant studies. The antiangiogenic mechanisms of above agents were described in **Table 2**.

### Characteristics of included studies

**Table 3** presented an overview of the patients' baseline characteristics in the 5 studies. The included studies showed a relatively uniform distribution of patients' characteristics at baseline. Treatment regimens and endpoints for the studies were showed in **Table 4**. **Table 5** provided methodological information that could potentially cause bias. Only 1 study included in this analysis was open-label [13]. The allocation concealment of randomization was presented in 2 studies [16, 18]. All the 5 included studies mentioned withdraw descriptions. It was important to note that there were differences in the exclusion criteria among the 5 studies. 2 studies excluded patients with substantial cardiovascular comorbidity [13, 17], while the other 3 studies did not [16, 18, 19].

The methods of measurements to review tumor responses were not uniform between the included studies in our analysis. Due to the non-radial growth pattern and inconsistent treatment response, it was difficult to measure MPM radiographically. Inconsistent methods of measurements would influence the final results of PFS and ORR in the clinical trials. The criteria of measurement methods for MPM in the trials were either the RECIST (Response Evaluation Criteria In Solid Tumours) or the newer modified RECIST (mRECIST). The RECIST

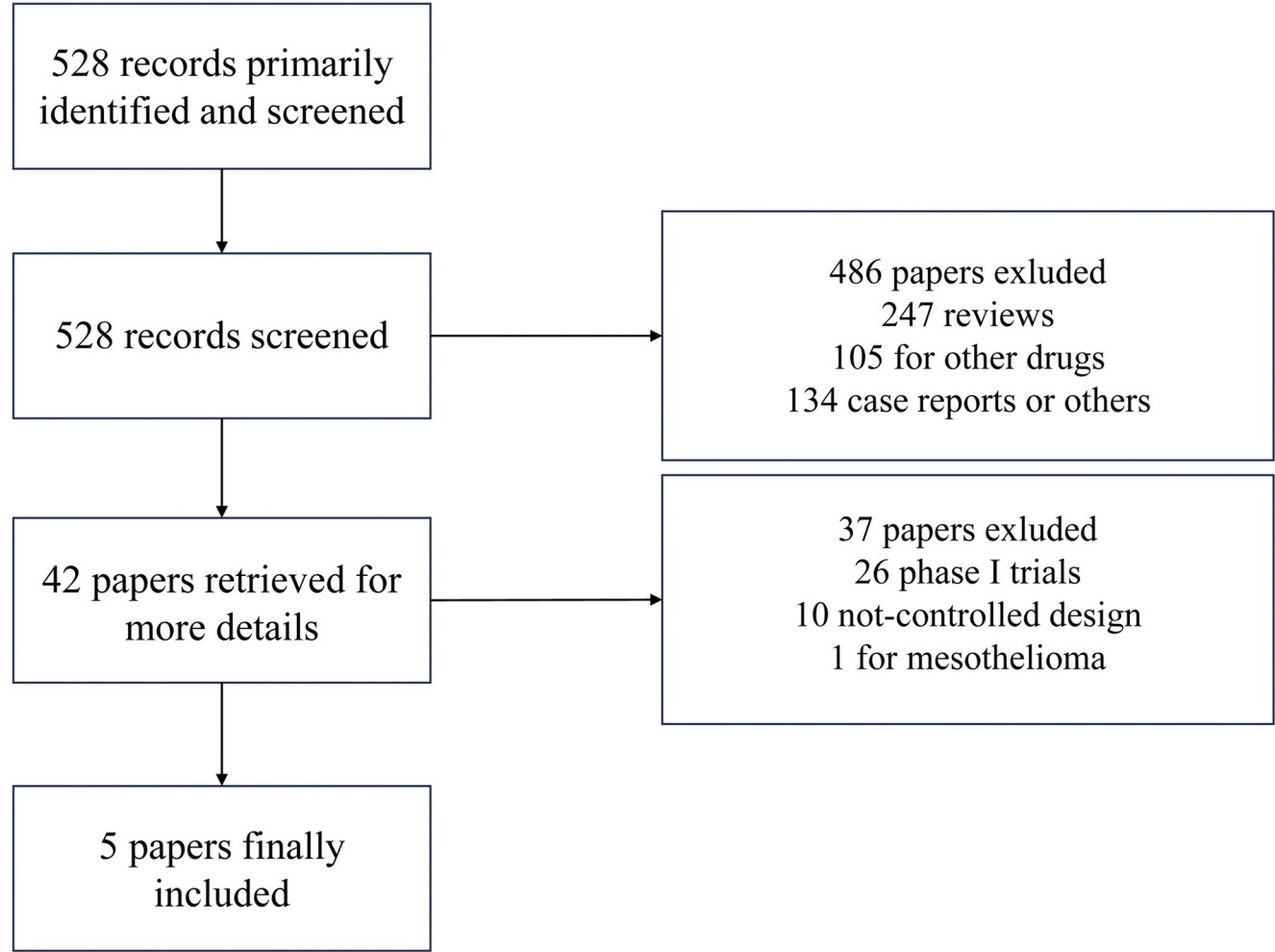

**Fig 1. Flowchart for identification and inclusion of studies for this analysis.**

criteria for MPM was to measure the long axis of the tumor to achieve the longest uni-dimensional measurement. The 2 longest diameters of the pleural tumor were used to calculate the total measurement. The mRECIST criteria was to measure tumor thickness perpendicular to the chest wall or mediastinum in two positions at three separate levels. The MPM measure was the sum of the six measurements [20]. Only 1 study used the RECIST criteria to evaluate tumor response [18]. Other 3 studies used the mRECIST criteria [13, 16, 19]. Tumor

**Table 2. Antiangiogenic mechanisms of different agents.**

| antiangiogenic agents | antiangiogenic mechanisms |
|---|---|
| bevacizumab | monoclonal antibody inhibitor of VEGF |
| nintedanib | angiokinase inhibitor of VEGFR 1–3, PDGFR-α and β, FGFR 1–3 |
| cediranib | tyrosine kinase inhibitor of VEGFR and PDGFR |
| ramucirumab | monoclonal antibody inhibitor of VEGFR 2 |

VEGF: vascular endothelial growth factor. VEGFR: vascular endothelial growth factor receptor. PDGFR: platelet-derived growth factor receptor. FGFR: fibroblast growth factor receptor.

**Table 3. Patients' characteristics of different arms in the included studies.**

| References | Histology | Treatment Arms | Epithelioid (%) | Patients Enrolled | Male (%) | ECOG ≤1 or KPS ≥70 (%) | Median age (years) |
|---|---|---|---|---|---|---|---|
| Gerard et al. 2016 [13] | malignant pleural mesothelioma | combination | 80 | 223 | 75 | 97 | 65.7 |
| | | chemotherapy | 81 | 225 | 76 | 96 | 65.6 |
| Federica et al. 2017 [19] | malignant pleural mesothelioma | combination | 89 | 44 | 77 | 100 | 68 |
| | | chemotherapy | 88 | 43 | 81 | 100 | 66 |
| Anne S et al. 2019 [17] | malignant pleural mesothelioma | combination | 76 | 45 | 84 | 93 | 72 |
| | | chemotherapy | 74 | 47 | 85 | 94 | 72 |
| Giorgio et al. 2019 [16] | malignant pleural mesothelioma | combination | 96 | 229 | 72 | 100 | 66 |
| | | chemotherapy | 97 | 229 | 74 | 100 | 66 |
| Carmine et al. 2021 [18] | malignant pleural mesothelioma | combination | 85 | 82 | 74 | 99 | 69 |
| | | chemotherapy | 86 | 83 | 74 | 99 | 69 |

ECOG: Eastern Cooperative Oncology Group. KPS: Karnofsky Performance Status.

measurements were determined by both RECIST and mRECIST in the study conducted by Anne et al [17]. Tumor response assessment was performed using computed tomography (CT) in 4 studies [13, 17–19]. Additionally, Giorgio et al conducted their study using either CT or magnetic resonance imaging (MRI) [16].

OS was defined as the duration from randomization to death from any cause. PFS was defined as the duration from randomization to disease progression or death from any cause. Two studies did not provide the detailed explanations of OS and PFS [16, 17]. The frequency of imaging test to assess tumor response varied among the included studies. Imaging scans were performed every 6 weeks in 3 studies [16, 17, 19] and every 9 weeks in 2 studies [13, 18]. Only 1 study explicitly stated that the imaging tests had been reviewed by an independent review committee [16].

## Overall survival

The impact of antiangiogenic agents on OS was extracted directly from the 5 included studies. In the analysis, the combination of antiangiogenic agents and chemotherapy was associated

**Table 4. Regimens and endpoints of included studies.**

| References | Regimens (per arm) | Interventions | Primary endpoint | Line |
|---|---|---|---|---|
| Gerard et al. 2016 [13] | Pem+CDDP+Bev | Arm A: Pem 500 mg/m² iv d1, CDDP 75 mg/m² iv d1, Bev 15 mg/kg iv d1, q3w. | OS | First |
| | Pem+CDDP | Arm B: Pem 500 mg/m² iv d1, CDDP 75 mg/m² iv d1, q3w. | | |
| Federica et al. 2017 [19] | Pem+CDDP+Nin | Arm A: Pem 500 mg/m² iv d1, CDDP 75 mg/m² iv d1, Nin 400 mg/d po d2-21, q3w. | PFS | First |
| | Pem+CDDP | Arm B: Pem 500 mg/m² iv d1, CDDP 75 mg/m² iv d1, q3w. | | |
| Anne S et al. 2019 [17] | Pem+CDDP+Ced | Arm A: Pem 500 mg/m² iv d1, CDDP 75 mg/m² iv d1, Ced 20 mg/d po d1-21, q3w. | PFS | First |
| | Pem+CDDP | Arm B: Pem 500 mg/m² iv d1, CDDP 75 mg/m² iv d1, q3w. | | |
| Giorgio et al. 2019 [16] | Pem+CDDP+Nin | Arm A: Pem 500 mg/m² iv d1, CDDP 75 mg/m² iv d1, Nin 400 mg/d po d2-21, q3w. | PFS | First |
| | Pem+CDDP | Arm B: Pem 500 mg/m² iv d1, CDDP 75 mg/m² iv d1, q3w. | | |
| Carmine et al. 2021 [18] | Gem+Ram | Arm A: Gem 1000mg/m² iv d1 and d8, Ram 10mg/kg iv d1, q3w. | OS | Second |
| | Gem | Arm B: Gem 1000mg/m² iv d1 and d8, q3w. | | |

Pem: pemetrexed. CDDP: cisplatin. Bev: bevacizumab. Nin: nintedanib. Ced: cediranib. Gem: gemcitabine. Ram: ramucirumab. OS: overall survival. PFS: progression free survival.

**Table 5. Methodological details which might cause bias of included studies.**

| References | Phase | Random | Blind | Randomization description | Concealment description | Withdraw description | ITT analysis | Multicenter | Jadad score |
|---|---|---|---|---|---|---|---|---|---|
| Gerard et al. 2016 [13] | III | Yes | No | Yes | No | Yes | Yes | Yes | 3 |
| Federica et al. 2017 [19] | II | Yes | Yes | NC | NC | Yes | Yes | Yes | 3 |
| Anne S et al. 2019 [17] | II | Yes | Yes | NC | NC | Yes | Yes | No | 3 |
| Giorgio et al. 2019 [16] | III | Yes | Yes | Yes | Yes | Yes | Yes | Yes | 5 |
| Carmine et al. 2021 [18] | III | Yes | Yes | Yes | Yes | Yes | Yes | Yes | 5 |

NC: no clear. ITT: intend-to-treat.

with longer OS compared with chemotherapy alone [HR 0.79 (0.71–0.89), p<0.0001] (**Fig 2**), without significant heterogeneity between studies ($I^2$ = 31%, p = 0.22).

The patients in the 5 included studies were divided into epithelioid and non-epithelioid subgroups based on the pathological types. All the 5 included studies reported the data for epithelioid subgroup. 4 included studies reported available data for non-epithelioid subgroup [13, 16–18]. The analysis showed that the combination therapy was associated with longer survival in the epithelioid subgroup [HR 0.84 (0.72–0.97), p = 0.02] ($I^2$ = 0%, p = 0.73) (**Fig 3**), but not in the non-epithelioid subgroup [HR 0.73 (0.53–1.00), p = 0.05] ($I^2$ = 66%, p = 0.03) (**Fig 4**). Significant heterogeneity was found in the non-epithelioid subgroup analysis. When random-effect model was applied, no advantage on OS in the non-epithelioid subgroup was observed [HR 0.78 (0.43–1.40), p = 0.40] (**S1 Fig**).

Of the 5 studies, 4 studies included patients that had not received any previous systematic therapies, and only the study conducted by Carmine et al included patients that had undergone first-line treatment. The superior OS advantage was observed in the first-line setting [HR 0.85 (0.74–0.99), p = 0.04], without significant heterogeneity ($I^2$ = 13%, p = 0.33) (**Fig 5**).

## Progression free survival

All the 5 included studies reported the data of PFS. 4 studies analyzed PFS according to the mRECIST criteria [13, 16, 17, 19], while the study conducted by Carmine et al used the RECIST criteria [18]. The addition of antiangiogenic agents to chemotherapy improved PFS

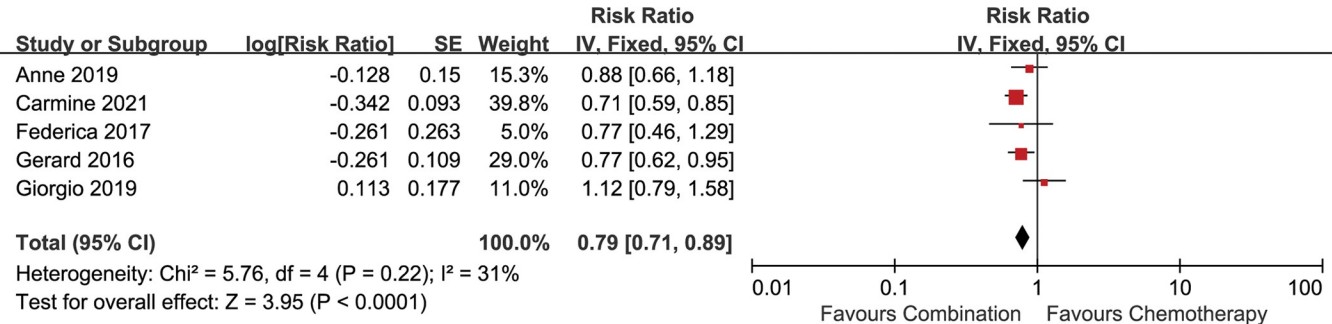

**Fig 2. Comparison of OS between addition of antiangiogenic agents to chemotherapy and chemotherapy alone.** SE: standard error. CI: confidence interval. IV: inverse variance.

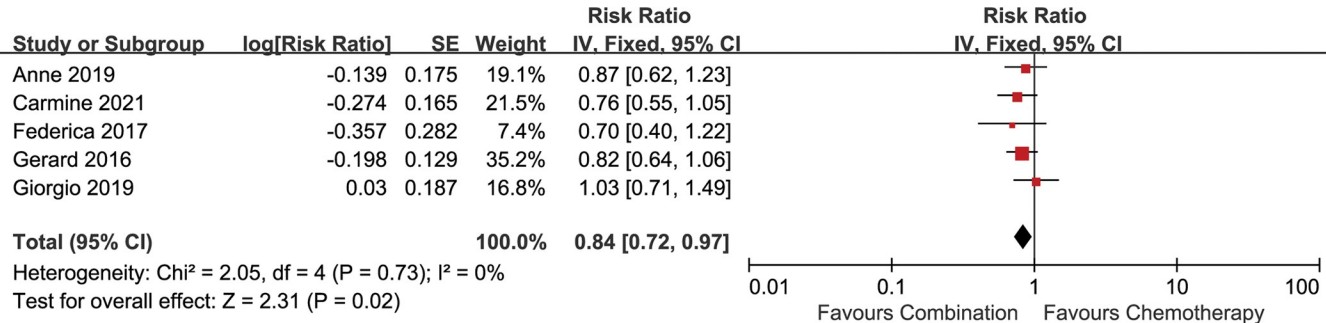

**Fig 3. Comparison of OS between addition of antiangiogenic agents to chemotherapy and chemotherapy alone in the epithelioid subgroup.** SE: standard error. CI: confidence interval. IV: inverse variance.

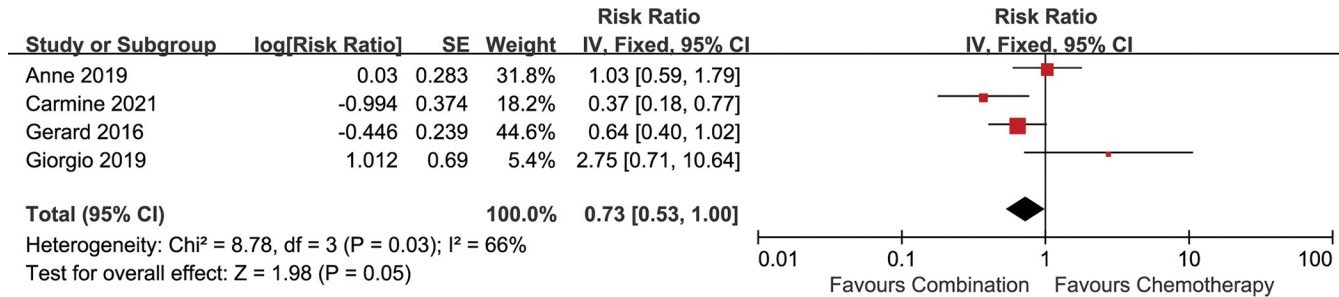

**Fig 4. Comparison of OS between addition of antiangiogenic agents to chemotherapy and chemotherapy alone in the non-epithelioid subgroup.** SE: standard error. CI: confidence interval. IV: inverse variance.

compared with chemotherapy alone [HR 0.75 (0.68–0.84), p<0.00001], with significant heterogeneity among studies ($I^2 = 66\%$, p = 0.02) (**Fig 6**). When random-effect model was applied, the advantage on PFS was also observed [HR 0.75 (0.62–0.91), p = 0.003] (**S2 Fig**).

To ensure consistency in the criteria for evaluating efficacy, we performed analysis of PFS by excluding the study conducted by Carmine et al. The analysis showed that addition of antiangiogenic agents was still associated with superior PFS [HR 0.73 (0.64–0.84), p<0.00001], with significant heterogeneity ($I^2 = 73\%$, p = 0.01) (**Fig 7**). The superiority of PFS was also observed when random-effect model was used [HR 0.73 (0.56–0.96), p = 0.02] (**S3 Fig**). The

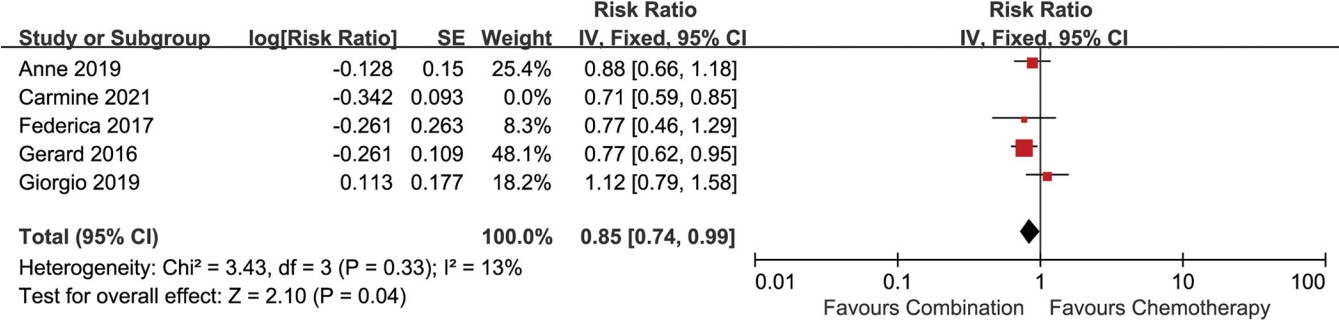

**Fig 5. Comparison of OS between addition of antiangiogenic agents to chemotherapy and chemotherapy alone in the first-line setting.** SE: standard error. CI: confidence interval. IV: inverse variance.

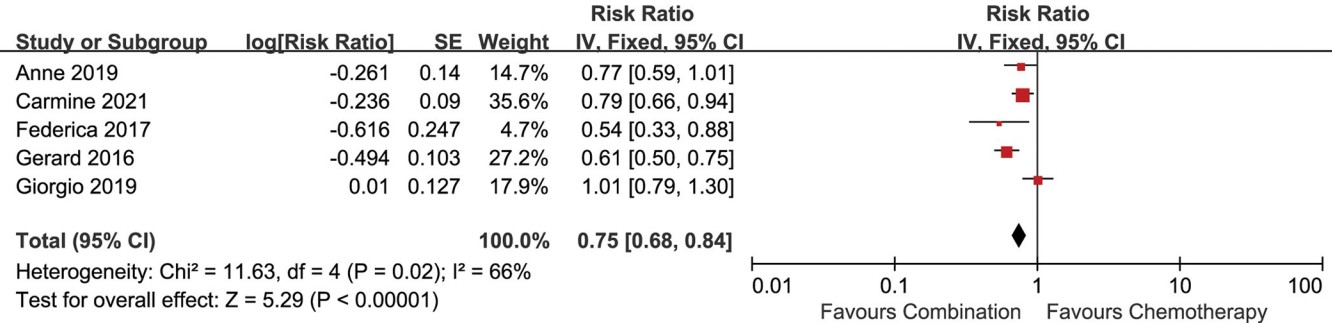

**Fig 6. Comparison of PFS between addition of antiangiogenic agents to chemotherapy and chemotherapy alone.** SE: standard error. CI: confidence interval. IV: inverse variance.

subgroup analysis of PFS in first-line setting was the same as the above result when excluding the study conducted by Carmine et al.

## Overall response rate

4 included studies described the impact of combination therapy on ORR [16, 17–19]. The analysis showed no significant difference between two arms [HR 1.13 (0.95–1.35), p = 0.18], with moderate heterogeneity ($I^2$ = 49%, p = 0.12) (**Fig 8**).

Subgroup analysis showed that no advantage of ORR was observed for antiangiogenic agents combined with chemotherapy in the first-line setting [HR 1.16 (0.97–1.40), p = 0.10], with significant heterogeneity ($I^2$ = 58%, p = 0.09) (**Fig 9**). When applying the random-effect model, no ORR advantage was observed [HR 1.30 (0.89–1.89), p = 0.17] (**S4 Fig**).

## Toxicities

Reported toxicities with grade≥3 for the combination therapy were assessed in our analysis. We evaluated the toxicities of hypertension, thromboembolism, proteinuria and bleeding events primarily caused by antiangiogenic agents, and other common toxicities associated with conventional chemotherapy. Only 1 study reported the toxicity of proteinuria, including 1 case [18]. Bleeding events were rare, occurring in only 3 patients receiving combination therapy in 2 studies [13, 17]. There were more incidences of grade≥3 hypertension [RR 28.11 (8.07–97.89), p<0.00001], neutropenia [RR 1.20 (1.02–1.41), p = 0.03] and diarrhea [RR 2.50 (1.22–5.15), p = 0.01] when combining antiangiogenic agents with chemotherapy.

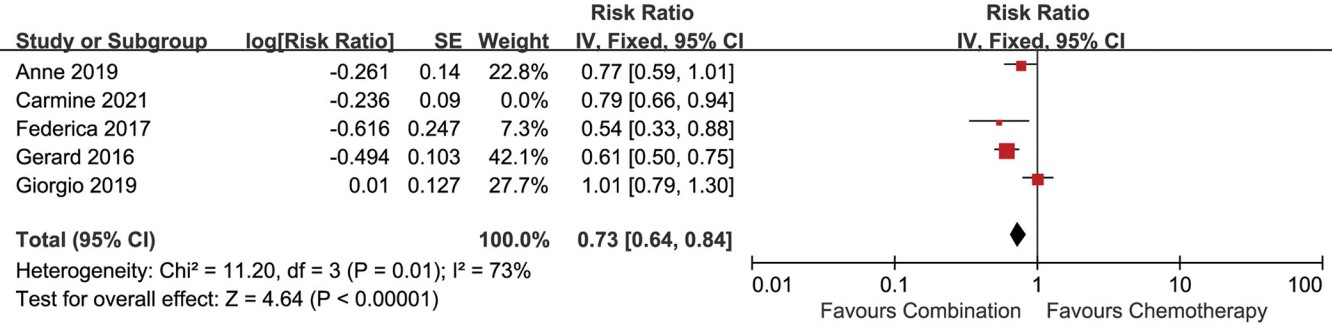

**Fig 7. Comparison of PFS between addition of antiangiogenic agents to chemotherapy and chemotherapy alone according to the mRECIST criteria.** SE: standard error. CI: confidence interval. IV: inverse variance.

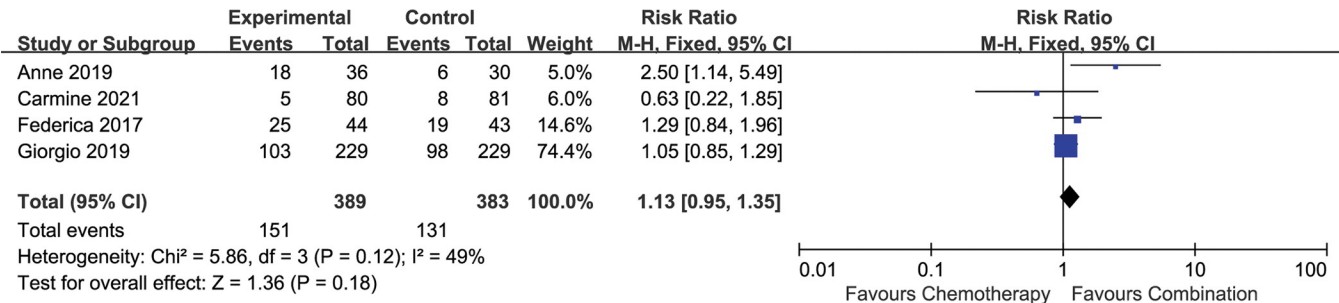

**Fig 8. Comparison of ORR between addition of antiangiogenic agents to chemotherapy and chemotherapy alone.** CI: confidence interval. M-H: Mantel-Haenszel.

Interestingly, the addition the antiangiogenic agents to chemotherapy reduced the incidence of grade≥3 anemia [RR 0.51 (0.35–0.73), p = 0.0003]. Equivalent frequencies of thromboembolism, thrombocytopenia, fatigue, nausea and vomiting were found between the two arms (**Fig 10**).

## Publication bias

We utilized a highly sensitive search strategy to select relevant studies, thereby minimizing the possibility of publication bias. And the collection of papers was strictly in accordance with the inclusion criteria. Publication bias was determined using funnel plots. Egger's test provided statistical evidence for the symmetry of the funnel plot. There was no apparent publication bias detected in our analysis (**Fig 11**). Egger's test found no apparent evidence of bias in OS (p = 0.27) and PFS (p = 0.78).

## Discussion

Regimen of pemetrexed plus cisplatin, approved by Food and Drug Administration (FDA), had been considered as the standard care for first-line treatment in unresectable MPM. It had been proven to be quite challenging in making significant improvements in systemic therapy for advanced MPM. Antiangiogenic agents had been combined with the current standard of care in the first or second-line setting. However, the approach had yielded mixed results. Our analysis showed that addition of antiangiogenic agents to chemotherapy led to improvements on OS and PFS, with a tolerable toxicity profile.

Biomarkers that could reliably predict treatment response were of high clinical relevance. Increased platelet count could secrete angiogenic factors such as VEGF and PDGF, or directly

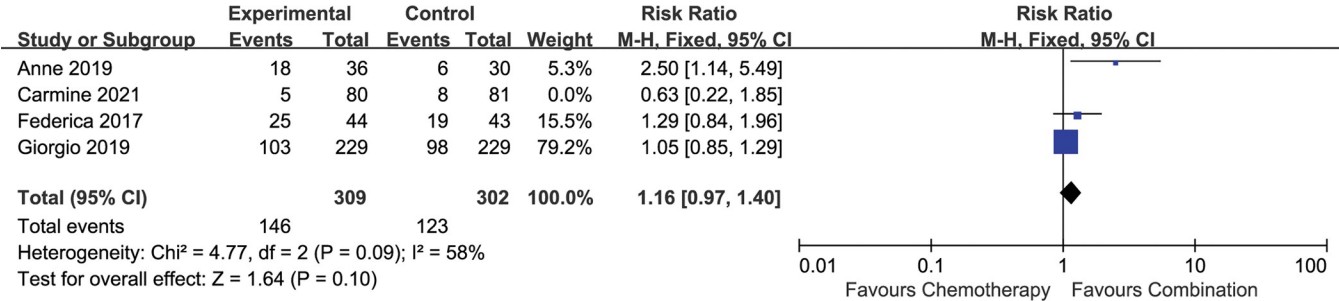

**Fig 9. Comparison of ORR between addition of antiangiogenic agents to chemotherapy and chemotherapy alone in the first-line setting.** CI: confidence interval. M-H: Mantel-Haenszel.

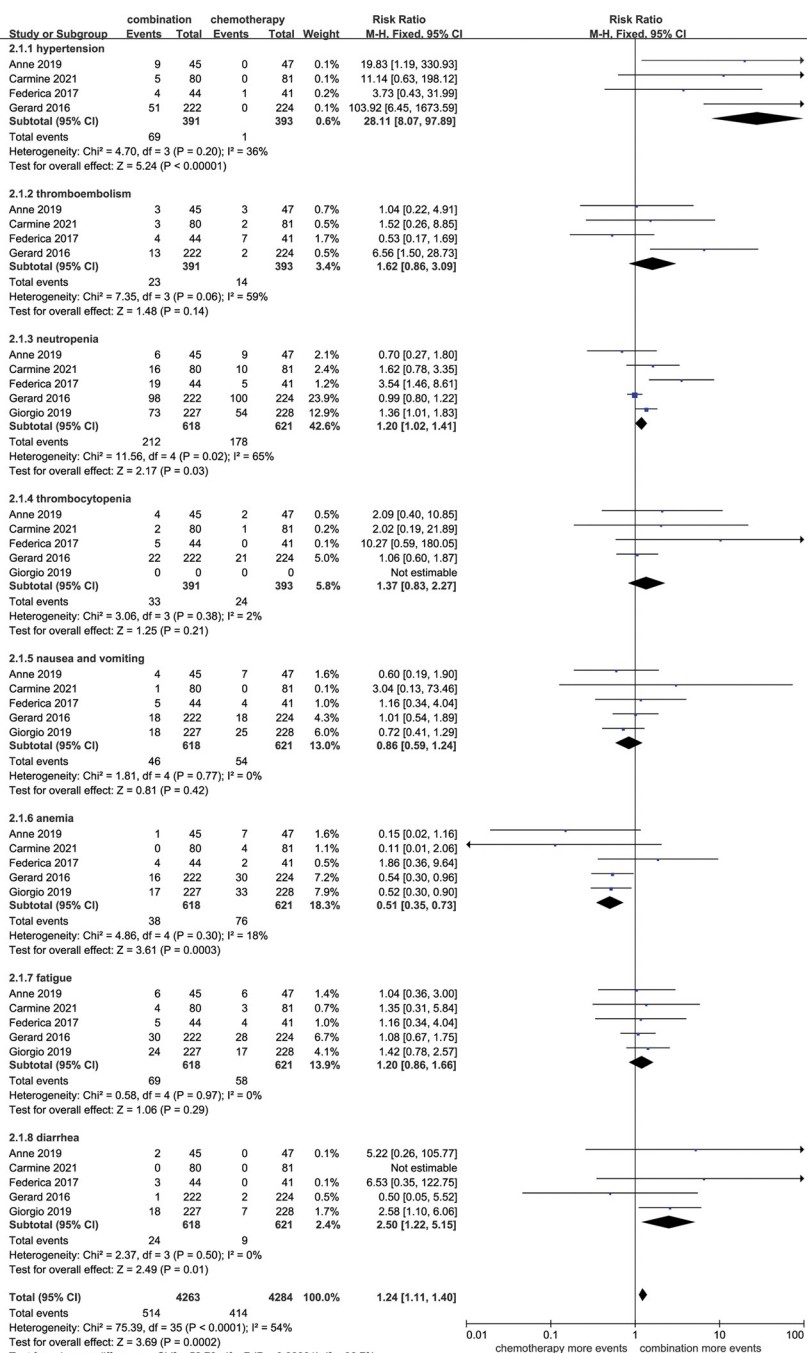

**Fig 10. Comparison of toxicities between addition of antiangiogenic agents to chemotherapy and chemotherapy alone.** CI: confidence interval. M-H: Mantel-Haenszel.

stimulate the progression of angiogenesis to augment tumor growth in many types of solid tumors [21]. Higher platelet count had been considered to be a well-established predictive factor with poor OS and shorter PFS in MPM [22]. Furthermore, the increased expressions of VEGF and VEGFR in serum and tissues were associated with significantly poorer prognoses in patients with MPM [23]. Subgroup analyses according to baseline platelet count were only reported in two included studies in our analysis. As a result, no predictive biomarker that

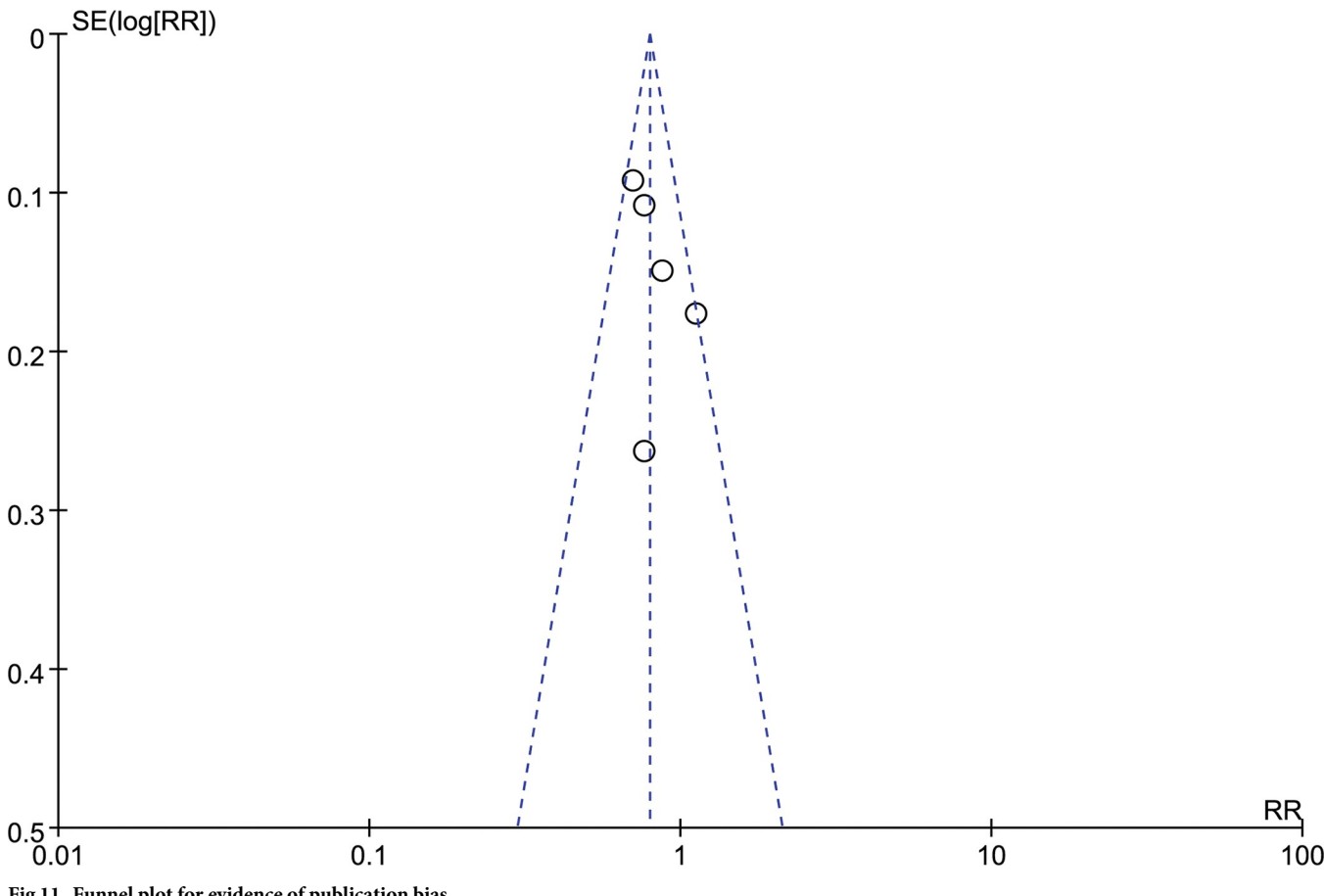

**Fig 11. Funnel plot for evidence of publication bias.**

could guide treatment and improve patient selection was identified in our analysis. Additional studies should be warranted in future to identify the potential biomarkers, such as platelet count or VEGF concentration to inform treatment decision and patient selection.

In our analysis, no advantage on ORR was observed by adding antiangiogenic agents to chemotherapy in advanced MPM. Due to the complex morphology of MPM, response evaluation could easily cause high variability among investigators. Endpoints that should be assessed radiographically, such as PFS, ORR and duration of response, were challenging in trials for the pleural-based disease. Although mRECIST for mesothelioma was widely used in clinical trials, ability of the criteria to uniformly evaluate tumor response remained limited. The improvement on OS might be the preferred endpoint for definitive clinical trials in advanced MPM [24]. More studies should focus on improving the critical value of response classification and developing innovative alternative endpoints in this special pleural-based disease.

Several studies had reported the role of immunotherapy on MPM. Nivolumab monotherapy or nivolumab combined with ipilimumab both showed promising 12-week disease control rate (40% and 52%) in relapsed patients with MPM in the phase II randomized IFCT-1501 MAPS2 study [25]. The CheckMate 743 study was a phase III randomized controlled trial compared the efficacy of nivolumab plus ipilimumab with standard chemotherapy in patients with advanced MPM. Compared with chemotherapy, nivolumab plus ipilimumab enhanced OS (18.1 vs 14.1 months) in the first-line setting [26]. These studies indicated that immunotherapy could bring clinical benefits in patients with advanced MPM. Nivolumab and

ipilimumab might be considered as the new standard of treatment in the near future. The combination of antiangiogenic agents and immune checkpoint inhibitors was considered to be synergistic in preclinical studies [27]. The IMPower 150 study was a phase III randomized controlled trial evaluated atezolizumab plus bevacizumab plus chemotherapy in patients with metastatic non-small-cell lung cancer. Compared with paclitaxel-carboplatin-bevacizumab, the addition of atezolizumab improved OS (19.2 months vs 14.7 months) [28]. More clinical studies should be needed to assess the addition of immune checkpoint inhibitors to antiangiogenics and chemotherapy in advanced MPM.

The following were some limitations existed in our analysis. Firstly, due to the small number of studies included, the power of the analysis was limited. Secondly, the combination of phase II and phase III trials was another limitation. OS was not always the primary endpoint in phase II trials. As a result, only two studies included in this analysis used OS as primary endpoint. The follow-up period in study using PFS as primary endpoint might be shorter than that using OS, which led to the non-equivalent data maturity between studies. Thirdly, clinical benefits of combining antiangiogenic agents and chemotherapy might be distinct in patients with different ages and general conditions. More subgroup analyses would be needed to make the conclusion stronger. But the dividing points of ages and general conditions for subgroup analyses were not uniform between studies, which acted as obstacles for further exploration. Fourthly, it was worth noting that hypoxia had the potential to enhance aggressiveness in malignant mesothelioma cells, thereby influenced various biological and molecular aspects. Previous in vitro experiments had substantiated the fact that hypoxia could stimulate clonogenicity, migration, invasion, and drug resistance to cisplatin in malignant mesothelioma cells [29]. However, none of the 5 studies included in this analysis provided information regarding the number of patients who underwent oxygen therapy, which could potentially impact the final outcomes of OS and PFS. Fifthly, among the 5 studies included, specific statistics on the number of patients who underwent pleural biopsy (percutaneous puncture or thoracoscopic surgery) were not provided. Most patients underwent thoracoscopic surgery would receive pleurodesis, which could potentially impact the prognosis. Finally, 2 studies excluded patients with significant cardiovascular comorbidity, potentially resulting in a study population with a more favorable prognosis. However, it was worth noting that the contraindications for the use of antiangiogenic agents were not strictly adhered to in a real-life setting, which might have a significant impact on the outcome, potentially leading to increased mortality. These factors should be taken into account in future clinical trials.

In conclusion, the combination of antiangiogenic agents and chemotherapy showed advantage compared with chemotherapy alone in patients with advanced MPM. Future studies should focus on identifying the potential biomarkers to distinguish patients who would most likely benefit from the combination regimen. Furthermore, prospective trials should be warranted to determine the synergetic effect of adding immune checkpoint inhibitors to antiangiogenic agents and chemotherapy.

## Supporting information

**S1 Appendix. PRISMA flow diagram.**
(DOC)

**S1 Table. PRISMA checklist.**
(DOCX)

**S1 Fig. Comparison of OS between addition of antiangiogenic agents to chemotherapy and chemotherapy alone in the non-epithelioid subgroup using random-effect model.** SE:

standard error. CI: confidence interval. IV: inverse variance.
(TIF)

**S2 Fig. Comparison of PFS between addition of antiangiogenic agents to chemotherapy and chemotherapy alone using random-effect model.** SE: standard error. CI: confidence interval. IV: inverse variance.
(TIF)

**S3 Fig. Comparison of PFS between addition of antiangiogenic agents to chemotherapy and chemotherapy alone according to the mRECIST criteria using random-effect model.** SE: standard error. CI: confidence interval. IV: inverse variance.
(TIF)

**S4 Fig. Comparison of ORR between addition of antiangiogenic agents to chemotherapy and chemotherapy alone in the first-line setting using random-effect model.** CI: confidence interval. M-H: Mantel-Haenszel.
(TIF)

## Acknowledgments

We thank Haimei Wang for her helpful advice in searching strategy.

## Author Contributions

**Conceptualization:** Wei Tian.

**Data curation:** Wei Tian, Daidi Fu, Xiao Ma.

**Methodology:** Wei Tian, Daidi Fu, Rui Wang.

**Resources:** Qian Guo.

**Software:** Wei Tian, Qian Guo.

**Writing – original draft:** Wei Tian.

**Writing – review & editing:** Daidi Fu, Xiao Ma, Rui Wang.

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
