## [Decision Letter · Decision Letter 0]

9 Nov 2023

PONE-D-23-29143Efficacy and safety profile of combining antiangiogenic agents with chemotherapy in patients with advanced malignant pleural mesothelioma: a systematic review and meta-analysis of randomized controlled trialsPLOS ONE

Dear Dr. Wang,

Thank you for submitting your manuscript to PLOS ONE. After careful consideration, we feel that it has merit but does not fully meet PLOS ONE’s publication criteria as it currently stands. Therefore, we invite you to submit a revised version of the manuscript that addresses the points raised during the review process.

We look forward to receiving your revised manuscript.

Kind regards,

Jun Hyeok Lim, M.D.

Academic Editor

PLOS ONE

Journal Requirements:

Reviewers' comments:

Reviewer's Responses to Questions

**Comments to the Author**

1. Is the manuscript technically sound, and do the data support the conclusions?

Reviewer #1: Yes

2. Has the statistical analysis been performed appropriately and rigorously? 

Reviewer #1: I Don't Know

3. Have the authors made all data underlying the findings in their manuscript fully available?

Reviewer #1: Yes

4. Is the manuscript presented in an intelligible fashion and written in standard English?

Reviewer #1: Yes

5. Review Comments to the Author

Reviewer #1: The authors have done an incredible job in this research subject especially so since it is a rare malignancy and extensive efforts to identify bio markers have been unsuccessful.

I have a few questions for the authors though-

1) What kind of imaging modality was used in the patients? Was it a CECT or MRI?

2) How many patients were hypoxic and on oxygen therapy since hypoxia itself is a tumor promoting state because it reduces apoptosis, increases tyrosine kinase activity, increases micro invasion and metastasis and all these could affect the outcome of the patients?

3)How many of the patients underwent a VATS as a diagnostic modality because VATS can cause pleurodesis in around 90% of times and do the authors think that this could affect the outcome of the study?

4) Were patients with cardiovascular problems excluded from the study because that again could have a significant impact on the outcome in terms of increased mortality?

Thank you.

6. PLOS authors have the option to publish the peer review history of their article (what does this mean?). If published, this will include your full peer review and any attached files.

Reviewer #1: No

---

## [Author Response · Author response to Decision Letter 0]

22 Nov 2023

Reply for manuscript “Efficacy and safety profile of combining antiangiogenic agents with chemotherapy in patients with advanced malignant pleural mesothelioma: a systematic review and meta-analysis of randomized controlled trials”

 Reply to editor:

Dear editor:

Thank you for giving us the chance to submit the revised manuscript. We sincerely appreciate the reviewer’s constructive comments. Taking these valuable suggestions into consideration, we have made extensive revisions to the manuscript. This includes correcting errors and providing additional materials to improve its quality. Furthermore, we have ensured that the manuscript meets all of PLOS ONE's style requirements.

Answer: We have carefully reviewed our manuscript to ensure that it meets PLOS ONE's style requirements.

Answer: We appreciate your kind reminder, but we have not made arrangements to deposit the raw data in a repository at this time.

Answer: We revised the Data Availability Statement in the cover letter. All relevant data are within the paper and its Supporting information files.

Answer: We have included captions for Supporting Information at the end of our manuscript. Line 338-340.

Answer: We have reviewed our reference list to ensure it meets the criteria outlined in the PLOS ONE guidelines for References. In response to the reviewer's question, we have included an additional paper [29] in the discussion section. Line 440-442.

Reply to reviewer:

Dear reviewer:

Thank you for dedicating your time and providing valuable feedback on our manuscript. We truly value your insightful comments, as they greatly contribute to the enhancement of our manuscript. We have thoroughly analyzed and addressed each comment individually, denoting the line number in the revised manuscript to track the alterations made. Our intention is to ensure that the revised version aligns more effectively with your expectations. If necessary, we are willing to make additional amendments to the manuscript to further enhance its quality.

Reply to reviewer 1:

Reviewer #1: The authors have done an incredible job in this research subject especially so since it is a rare malignancy and extensive efforts to identify bio markers have been unsuccessful.

I have a few questions for the authors though-

1)What kind of imaging modality was used in the patients? Was it a CECT or MRI?

Answer: In the revised manuscript, we included an illustration about the radiographical method used to assess tumor response in the 5 included studies. Line 178-180.

2)How many patients were hypoxic and on oxygen therapy since hypoxia itself is a tumor promoting state because it reduces apoptosis, increases tyrosine kinase activity, increases micro invasion and metastasis and all these could affect the outcome of the patients?

Answer: None of the 5 studies included in this analysis provided information regarding the number of patients who underwent oxygen therapy, which could potentially impact the final outcomes of OS and PFS. We have acknowledged this limitation in our manuscript. Line 315-320.

3)How many of the patients underwent a VATS as a diagnostic modality because VATS can cause pleurodesis in around 90% of times and do the authors think that this could affect the outcome of the study?

Answer: None of the 5 studies mentioned the number of patients underwent a VATS as a diagnostic modality. Specific statistics on the number of patients who underwent pleural biopsy (percutaneous puncture or thoracoscopic surgery) were not provided. This might have impacted the prognosis of patients. We added the illustration in the limitation part. Line 320-323.

4)Were patients with cardiovascular problems excluded from the study because that again could have a significant impact on the outcome in terms of increased mortality?

Answer: There were variations in the exclusion criteria among the 5 studies. 2 studies excluded patients with significant cardiovascular comorbidity, whereas the remaining 3 studies did not. We acknowledged this difference as a limitation in our manuscript. Line 323-327.

---

## [Editor Report · Decision Letter 1]

29 Nov 2023

Efficacy and safety profile of combining antiangiogenic agents with chemotherapy in patients with advanced malignant pleural mesothelioma: a systematic review and meta-analysis of randomized controlled trials

PONE-D-23-29143R1

Dear Dr. Wang,

We’re pleased to inform you that your manuscript has been judged scientifically suitable for publication and will be formally accepted for publication once it meets all outstanding technical requirements.

Kind regards,

Jun Hyeok Lim, M.D.

Academic Editor

PLOS ONE

---

## [Editor Report · Acceptance letter]

12 Dec 2023

PONE-D-23-29143R1 

Efficacy and safety profile of combining antiangiogenic agents with chemotherapy in patients with advanced malignant pleural mesothelioma: a systematic review and meta-analysis of randomized controlled trials 

Dear Dr. Wang:

I'm pleased to inform you that your manuscript has been deemed suitable for publication in PLOS ONE. Congratulations! Your manuscript is now with our production department. 

Kind regards, 

on behalf of

Dr. Jun Hyeok Lim 

Academic Editor

PLOS ONE